# “Understand the Way We Walk Our Life”: Indigenous Patients’ Experiences and Recommendations for Healthcare in the United States

**DOI:** 10.3390/ijerph22030445

**Published:** 2025-03-17

**Authors:** Melissa E. Lewis, Ivy Blackmore, Martina L. Kamaka, Sky Wildcat, Amber Anderson-Buettner, Elizabeth Modde, Laurelle Myhra, Jamie B. Smith, Antony L. Stately

**Affiliations:** 1Department of Family and Community Medicine, School of Medicine, University of Missouri, Columbia, MO 65211, USA; smithjami@umsystem.edu; 2Independent Researcher, St. Louis, MO 63124, USA; ivy.a.blackmore@gmail.com; 3Department of Native Hawaiian Health, John A. Burns School of Medicine, University of Hawaii at Manoa, Honolulu, HI 96822, USA; martinak@hawaii.edu; 4Independent Researcher, Salina, OK 74635, USA; skybwildcat@gmail.com; 5Department of Biostatistics and Epidemiology, Hudson College of Public Health, University of Oklahoma, Oklahoma City, OK 73104, USA; amber-s-anderson@ouhsc.edu; 6Department of Child and Adolescent Psychiatry, Brown University Health, Providence, RI 02906, USA; elizabeth_modde@brown.edu; 7Mino Bimaadiziwin Wellness Clinic, Minneapolis, MN 55404, USA; laurelle.myhra@redlakenation.org; 8Native American Community Clinic, Minneapolis, MN 55404, USA; astately@nacc-healthcare.org

**Keywords:** Indigenous, Indigenous research methods, Indigenous patients, culturally congruent care

## Abstract

Background: The quality of healthcare experiences for Indigenous communities is worse when compared to non-Indigenous patients. Bias and discrimination within healthcare systems relate to worsened care and worsened health outcomes for Indigenous patients. The purpose of this study was to learn about the experiences of Indigenous people within healthcare settings, as well as their viewpoints for improving healthcare delivery to this population. Methods: Indigenous research methods were employed in this study as clinic administrators and staff, elders, and Indigenous researchers collaborated on the study purpose, design, and analysis. Twenty Indigenous patients participated in one of four focus groups regarding their experiences with healthcare systems. Results: Seven main themes emerged, highlighting participants’ experiences during health encounters, in relation to healthcare systems, and Indigenous health beliefs. Participants discussed challenges and barriers in each area and offered recommendations for care delivery to this population. Conclusions: Participants in this study highlighted that biased care results in poor quality of healthcare delivery and that there are actionable steps that providers and systems of healthcare can take to reduce bias within healthcare systems. The provision of culturally congruent care is imperative in improving the health and well-being of Indigenous communities.

## 1. Introduction

Indigenous people worldwide experience significant health disparities [1,2,3,4,5,6]. Many of these disparities are due to the adverse impacts of colonization, including policies of genocide and forced assimilation. Colonization has resulted in the disruption of Indigenous culture and lifeways, as well as racism, historical and contemporary trauma, and discrimination and bias both at the personal and institutional level [1,2,3,4,5,6,7,8,9]. Indigenous people experience high rates of discrimination in a variety of healthcare settings. Compared to non-Hispanic white patients, Indigenous people are five to ten times more likely to experience discrimination [10,11] and are 2.6 times more likely to report racially biased maltreatment [12]. Discrimination in healthcare settings results in worsened care and health outcomes for Indigenous people [2,13]. Indigenous people across the United States have some of the highest rates of physical and mental health distress [14] and illness [15,16].

Bias, racism, and disrespect for health beliefs can result in the avoidance of care, which is linked to worse health outcomes for Indigenous people [2,10,12,13,14,17,18,19,20]. Nearly one in six Indigenous people do not seek healthcare for either their family members or themselves because of anticipated unfair treatment or discrimination [11]. Discrimination in healthcare settings carries serious health risks for Indigenous patients and can result not only the denial of proper treatment but also the avoidance of critical preventive visits, such as cancer screening [21]. Interestingly, healthcare professionals themselves note that Indigenous patients experience unwelcoming healthcare environments, stigma, and stereotyping, including clinical practice informed by racist beliefs [22,23]. Discriminatory hospital policies limit important cultural and spiritual practices, such as disallowing mothers to smudge their babies [24], which prevents Indigenous people from utilizing their own vital health knowledge systems [2,25,26].

The continued prevalence of discrimination in healthcare systems and the correlation with higher rates of adverse health outcomes highlights the need to better educate and train healthcare professionals in culturally appropriate care. For example, providers with a history of cultural competence or bias training had reduced Indigenous-specific bias, suggesting the benefits of these types of programs [27]. This study seeks to better understand the needs and experiences of Indigenous patients in order to guide critical foundational content for a much-needed, effective, practical, and overarching Indigenous health toolkit for healthcare providers.

## 2. Methods

The first step in facilitating effective change is understanding the perspectives and experiences of those receiving care. This manuscript describes a qualitative study rooted in Indigenous Research Methodology [28] that describes participants’ healthcare experiences and the changes they would like to see in the provision of care. A wide range of themes emerged from the analysis. This paper will provide an overview of the central themes and a synthesis of patient recommendations for healthcare improvement. The incorporation of these themes will inform the creation of an Indigenous health toolkit to guide healthcare providers in the provision of effective healthcare to Indigenous patients.

### 2.1. Indigenous Research Methodology

Indigenous Research Methodology (IRM) provided the foundation for this project, allowing for the utilization of Indigenous world views and addressing self-determination and decolonization, with a focus on the healthcare experiences of Indigenous people. IRM is utilized in the study design and project implementation and management. Cultivating relationships is a core Indigenous value and IRM explores how “validity can be informed through relationality [28]”. IRM, therefore, is grounded in relationship and “relationality”, which includes not only the interconnections between researchers themselves as well as community stakeholders but also extends more broadly to relationships with, and responsibility to, the community, environment, ancestors and future generations [29]. Key relations that were cultivated and nurtured in this project included Principal Investigator (PI), clinic chief executive officer (CEO), clinic staff, clinic elder’s advisory board, and patients, with the majority being Indigenous. A priority was made to hire Indigenous staff, and community meetings were held to discuss this research project while upholding local, Indigenous protocols [29].

### 2.2. Study Context

Participants were recruited from a US urban, upper midwestern health clinic that provides medical, behavioral, substance use, and dental services to more than 4500 individuals annually, 73% of whom are American Indian/Alaska Native (AI/AN). Approximately 75% of clinic patients fall below 100% of the federal poverty level. The majority of patients reside within a 2–5 mile radius of the clinic. Health insurance coverage for patients includes Medicaid (62%), Medicare (6%), private/commercial (12%), and uninsured (20%). There are approximately 40 providers healthcare providers at this clinic.

### 2.3. Data Collection

The project sought Indigenous patients who were willing to share their healthcare experiences and thoughts on needed services and training for providers. Purposive sampling strategies were used for recruitment. The study inclusion criteria were adult (age 18 or older), Indigenous, and seen at the clinic on the day of recruitment to ensure the recent use of healthcare services.

Participants who visited the clinic received flyers describing the study information during their appointment check-in. No staff reported disruptions of workflow due to research recruitment. If interested, participants contacted the study staff and a time and date for the focus group was arranged. Participants arrived at the clinic and were escorted to a conference room in a non-clinical area after business hours. Informed consent documents were presented, explained, and signed before focus group initiation. Twenty people in total completed informed consent forms and a brief demographic paper survey and participated in one of four focus groups in 2019. The focus groups were conducted by trained Indigenous graduate students who used a facilitator’s guide. Groups encompassed between 3 and 9 people in each group ranging in length from 30 to 110 min. Of note, during the focus groups, participants sometimes shared experiences from other healthcare clinics and we did not attempt to identify references to the site of recruitment versus other sites. All focus groups were audio recorded. Audio recordings were subsequently deidentified and transcribed by a third-party person prior to analysis. Focus group questions can be found in Appendix A. This study was approved by the University of Missouri Institutional Review Board.

## 3. Analysis

The utilization of both Reflective Lifeworld [30] and Narrative Research [31] approaches helped guide the analysis and interpretation of the focus group data. The aim of these approaches is to describe a phenomenon as experienced by the study participants and contextualize the emerging themes within a holistic global perspective [30,31]. This type of phenomenological analysis entails a movement between interpreting the stories within the whole, its parts, and against the whole to obtain to a harmonious understanding [28,30]. Once themes are identified, the focus returns to the whole to discover the full context and meaning of the stories. The approach of examining both the parts and the whole adds to the validity and objectivity of the process and facilitates the identification of a “red thread”, the information that links the experiences of individuals to broader social phenomena [32,33]. The Discussion section illustrates the “red thread” that emerged from the analysis [34] and helps to contextualize the detailed discussion of the focus group themes and subthemes afterwards.

A team of three Indigenous (enrolled citizens of tribal nations) researchers [M.L., A.A-B., and S.W.] together conducted a template analysis of the transcribed focus group content to arrive at the themes using the Reflective Lifeworld [30] and Narrative Research [31] approaches described above. Transcripts were read multiple times. The analysis team developed a list of themes to be applied during coding. This was followed by the joint coding of data, after which disagreements were discussed and a final consensus was reached for all coding decisions. Once this round of coding was completed, an initial template was defined that summarized the identified themes and further modifications were made as the analysis continued. The final template identified, categorized, and summarized the themes into main and subthemes. 

## 4. Results

### 4.1. Participant Demographics

Brief demographics were collected from 20 participants (Table 1). All were Indigenous people, with the majority identifying their tribal affiliation as falling into the broad tribal category of Anishinaabe (*n* = 15). The remaining five participants identified broadly as Lakota. Participants were a majority female (80%) with an average age of 46 years. Annual incomes ranged from USD 0 to USD 37,000 with USD 11,709 as the average household income.

Participants were asked what health issues, if any, they suffered from in the categories of physical health, mental health, and substance use/dependence. They could report as many as they liked. Over half of participants stated that they suffered from at least one physical diagnosis (e.g., diabetes, seizure disorder, fibromyalgia), the majority had a mental health diagnosis (e.g., depression, anxiety, post-traumatic stress disorder) and half reported suffering from substance abuse disorders (e.g., alcohol, methamphetamines, opiates). English was the most spoken language, but three participants reported that an Indigenous language was their first language.

The data that were collected were rich and broad, resulting in themes that were grouped into three main levels. The main themes were categorized as level one and include the health encounter, healthcare systems, and Indigenous knowledge and beliefs. Two of the main themes have further categorizations within their domains and constitute the second level of themes. Finally, third and fourth levels of themes stem from some of the second-level themes. The outline below describes these categorizations. Figures illustrating second-, third- and fourth-level themes can also be seen throughout the Results section.

The Healthcare Encounter
Ineffective health encountersEffective health encountersImprovements needed for healthcare encountersHealthcare SystemsD.Systemic and structural barriersE.Effective healthcare systemsF.Improvements needed for healthcare systemsIndigenous Knowledge and Beliefs

### 4.2. The Healthcare Encounter

Responses in this category of themes were focused on the patient–provider interaction.

#### 4.2.1. Ineffective Health Encounters

This theme is characterized by health encounters that were unwanted and resulted in negative outcomes. Encounters were described by participants as poor quality, and they felt providers were biased against them as Indigenous patients. Participants noted that ineffective health encounters included interactions in which providers did not explain health issues well, did not listen to or respect the patient and patient autonomy in decision-making, and appeared to lack knowledge about or interest in the patients’ lives and communities (Figure 1).

More explicitly, some participants encountered racism, stereotyping, and negative assumptions by providers, with participants reporting that these interactions resulted in inaccurate, inadequate, delayed, and poor-quality care, leading to worsening health conditions and increased experiences of pain.


*“… we’ve all had to deal (with) prejudice in one way or another, … I noticed that their body language is pretty obvious, and their tone of their voice is obvious that they’re not into seeing you…”*



*“I don’t feel I get treated fairly or get the same care as a non-Native would.”*


Participants noted that all people, including healthcare providers, are susceptible to learning negative stereotypes about Indigenous people and applying them in their occupation. Participants shared that bias is related to inadequate foundational educational systems that continue to depict Indigenous people as inferior.


*“And when we were all growing up…in the textbooks, we read bad things about Natives. So that’s how they are raised in school—that Native people were bad…today that’s what is still in the books.”*



*“Each tribe has their own culture and history and language and all. I ran into that a lot of times where, you know, if you, you tell them you’re Indian and all they think is war bonnets and horses.”*


Participants shared specific examples of bias they encountered from providers. Specifically, Table 2 provides an overview of three types of assumptions that participants experienced that negatively impacted the care they received. Consequently, some patients shared that they avoid Western medicine entirely to prevent these detrimental, racist encounters.

Biased assumptions toward Indigenous people resulted in a hierarchical approach to healthcare delivery. Participants discussed a lack of shared decision-making and trust in their knowledge as patients, with which they were unsatisfied.


*“I’m still a person and I still need to be included on the treatment and, no, you can’t make all the decisions…”*


Biased assumptions of Indigenous patients are false, are unpleasant and sometimes traumatic to patients, and result in medical errors and worsened health outcomes for Indigenous patients:


*“What I really disliked was that the doctor did not trust my mother’s intuition when it came to my child, and she pretty much dismissed what my thoughts and feelings were … and he went into a full-blown seizure after that. So, she (provider) was not hearing me. She just said that he was having fainting spells and I really sensed it was more than that, but she basically dismissed what I thought and that right there frustrates me…”*



*“It’s like when I kept having a bad headache here and the day I came, and they didn’t know I had a stroke.”*


#### 4.2.2. Effective Health Encounters

Participants shared personal examples of positive and effective healthcare encounters (Figure 2). Participants wanted healthcare that was free of bias and reflected Indigenous health beliefs and values. Patients valued having Indigenous-serving clinics that offered continuity of care. Patients also valued providers who were free of bias, were able to complete culturally appropriate assessments, utilized integrated care, and used skilled clinical decision-making in partnership with patients. This included verifying both health literacy and understanding. 

Patients shared that they wanted to feel truly cared about.


*“I have a cut on my toe and she listened to my lungs and she listened to my heart and, you know, I never get out of there without getting the once over, which is nice. She knows what’s going on.”*



*“…(The provider) makes sure that I understand the medications they’re putting me on, and how to take them or what I’m using this medication for.”*


Participants reported positive health outcomes when they received bias-free and effective care. They felt less pathologized and more comfortable sharing their personal life. Participants described that better access to medication improved medication adherence, follow-up calls resulted in improved treatment adherence, and provider praise led to increased confidence in completing treatment plans and improving diets.

#### 4.2.3. Improvements Needed for Healthcare Encounters

To provide quality care to Indigenous patients, participants felt that providers need to listen, learn, and have a basic knowledge of the culture, context, and community. Providers need to know the history of Indigenous people, particularly the history of colonization and historical trauma, as well as ongoing bias-related stress and trauma with its negative impacts on health (Figure 3). This is important for building trust and improving patient–provider interactions.


*“I come from a very Native background and we don’t really trust people and doctors; so, if they could somehow learn more about our different cultures and traditions and somehow make us feel more comfortable and like them.”*



*“…we have a hard time opening up. We’ve come from…histories with things that people tend to look down on. But to understand us is to know that those are a lot of times secondary symptoms [that are] deeper.”*


Participants believed that providers should understand the Indigenous cultures in the communities being served and acknowledge their own privilege.


*“You know, if they’re in your neighborhood, they should know the people that they’re dealing with, you know, instead of coming from way out in the suburbs and then, you know, coming here and then looking at you like you’re less than… They should know and experience the culture that they*
*’re dealing with.*
*”*


In addition to cultural and historical knowledge, participants also highlighted the importance of increasing patient agency in treatment decisions:


*“A lot of times, the ones that are getting the care, their [Indigenous patients] feedback, their input, they [providers] never considered it…a bunch of people that write stuff in books that never even lived it make our decisions for us…”*


### 4.3. Healthcare Systems

Second-level themes focused on broader topics such as clinic or systemic and structural-level issues that can shape and influence healthcare encounters and patients’ health outcomes.

#### 4.3.1. Structural Barriers

Participants described barriers to quality care in the form of policy restrictions or administrative issues and executive-level decisions. Other barriers mentioned included physical barriers such as transportation difficulties and a lack of resources, cultural barriers with regard to a lack of language access and Indigenous health interventions, as well as clinical encounter/visit-related barriers such as a lack of providers, leading to delays in care and billing issues. Participants noted the differences between health centers that pathologized illness versus those that provided the appropriate services needed to successfully treat the illness. Figure 4 illustrates the range of barriers that were identified.

Notably, structural barriers that affect the clinical encounter include scheduling, hiring, patient flow, privacy, language access, and provider training. Specific examples include the lack of Indigenous providers and staff, specialists, provider continuity, as well as lack of clinic support for Indigenous health beliefs, practices, and interventions, such as traditional Indigenous medicine and Indigenous trauma-informed care. Several participants expressed that they viewed non-Indigenous providers with less trust due to past negative experiences. For clinics that do not have Indigenous providers, participants wanted to see efforts by clinics to meaningfully include Indigenous culture and history and be more inclusive of Indigenous people.


*“We are the original people here at least on this continent; so, I would appreciate it if they would at least get to know our cultures and the background. They have special languages and interpreters for other people, why can’t they have the same thing.”*



*“I wish there was an option for that [traditional Indigenous medicine] instead of regular medicine. There’s just so many people hooked on stuff like opiates now because of that [Western medicine]. I wish they’d offer more traditional medicine you know, before they start pushing the pills on you.”*


The lack of healthcare resources, including equipment such as scanners, was another frequently mentioned barrier. When participants brought up the lack of equipment, several laughed, noting to the researchers that the insufficient resources allocated to Indigenous people is an inside joke and it is commonly known that you “don’t go to IHS (Indian Health Service) after May” because of resource shortages.

Another issue was seen in the discrediting of Indigenous-serving clinics themselves. One participant shared that their healthcare provider at a non-Indigenous-serving healthcare organization suggested that they not continue to receive services at an Indigenous-serving clinic because the provider believed it to be of poor quality. However,


*“…after he had read all my medical records, I went back for an appointment, and he actually apologized to me and told me that he could see that I had gotten very good care at the (Indigenous-serving healthcare center).”*


Structural barriers to healthcare overlap with the experiences of many other underserved communities, but it appears that Indigenous-specific discrimination plays a role in each domain. One participant highlighted how, after identifying as Indigenous, it was assumed they would not pay for care if billed later and they were told to pay up front:


*“I took my son over there and I gave them my tribal card and they looked at me and said they would not see my son unless I had the cash or a credit card to hand them.”*


Participants reported that structural issues resulted in a variety of negative outcomes, including emotional distress in response to poor treatment from staff, feeling dismissed and ignored, and increased negative feelings due to receiving care incongruent with their health beliefs, increased experiences of pain, delays in treatment leading to worsening health conditions, and inaccurate diagnosis and treatment.

#### 4.3.2. Effective Healthcare Systems

Participants highlighted the importance of addressing the cultural needs of Indigenous people and the provision of comprehensive care and social support. Some of these included straightforward decisions such as having transportation arranged, food available, or the health clinic conveniently located (Figure 5). Others involved broader practices such as the intentionality of choosing an Indigenous patient population and locating the clinic in the heart of an Indigenous community and neighborhood. Participants requested that healthcare centers, on an individual and community level, listen to patients’ needs and be an active member of their healthcare treatment team.

For example, hiring from within the community was seen as an extension of respecting the community, its needs, and its values. Furthermore, positive lived experience includes the clinic focusing on community health needs using recruiting methods and locations that are convenient for community members and providing respectful services. From simply seeing, “Natives coming in and out of here that lived in this community their whole time”, to intentionally making structural decisions that infuse culture into clinical practices, participants reported being happy with services in which Indigenous people and culture were central:


*“Just grateful to have (an Indigenous-serving clinic) here (in our community).”*


As highlighted by the following quote, respectful, culturally appropriate, integrated, and comprehensive care is critical for a positive experience with the healthcare system:


*“…I wanted to detox and then I saw the sign at the bus stop and it said to call. So, I came here and saw someone and they started treatment plan and changed my prescription that night… She (provider) acts, because she cares. She’s easy to talk to and you know I feel very satisfied and happy with the treatment plan. You know I was just another number at the other place. It’s great—they don’t treat me like a drug addict. And getting help and I see a counselor and see a therapist. I got to come here every day, but they got to arrange for me a cab every day and I’ve never had anyone offer to do that for me. I’ve never had anybody treat me that well. And they’re always feeding me, I mean they always have food here. I’m very satisfied with the treatment. With everybody, the therapist she’s always calling, it’s been great. I would recommend this to anybody. It’s great. I know they care about me.”*


The location for this study was cited as a positive example of an effective healthcare system. Participants who received care there noted how proud and grateful they were that the clinic was in their community and that seeing other Indigenous people from the community utilizing the clinic added to their comfort level in seeking care. The integration of the clinic into the fabric of the community was highlighted as important for building mutual trust and respect. Hiring people from within the community was an extension of respecting the community and its needs by providing appropriate clinical services and employment opportunities. The clinic also pinpointed community health needs, used recruiting methods and locations that were convenient for community members, and provided culturally respectful services.

#### 4.3.3. Improvements Needed for Healthcare Systems

Focus group participants had a clear vision of how healthcare institutions could better meet their needs and improve health outcomes for Indigenous people. An overview of the third-level themes is shown in Figure 6. Of primary importance was having culturally informed health teams of staff and providers that are free of bias and stereotyping and sensitive to trauma-informed care. Participants expressed that *“Natives want to go to somewhere that’s culturally involved, people that understand our people, our ways*”, and suggested the need for cultural training for new and existing employees at clinics that serve Indigenous people.

Analogous to clinical decision-making, participants expressed a desire for Indigenous patients to be included in the decision-making processes that clinics make at the administrative and executive levels. Patients are aware that healthcare policies and practices have significant impacts on the care they receive, as well as the well-being of their family and community.


*“I think that’s really important to have community inclusiveness at the decision-making table because we receive the services.”*



*“I think they could have more resources to do their job better.”*


### 4.4. Indigenous Knowledge and Beliefs

Participants want providers to understand the importance of traditional healing practices and holistic care. Participants would like to receive care that incorporates Indigenous health beliefs and practices with the provision of care that includes a culturally informed healthcare provider, a traditional healer on staff, access to and instructions for the use of traditional plant medicine, and access to traditional healing ceremonies (Figure 7). The types of integration of Indigenous health beliefs included wanting providers to be familiar with their culture, greeting them with their Indigenous language, and accessing only Traditional Indigenous Medicine without Western medicine.

Participants noted that infusing culture and community traditions into clinical practices, such as laughter and sharing a meal, improved the level of comfort participants experience in the clinical setting.


*“As you can tell…we do use a lot of laughter to be more comfortable, that’s what we do at a lot of our ceremonies at home, we gather and have meals and joke and laugh with one another to get comfortable… and that’s where we get our community togetherness…”*


Finally, respect for and inclusion of family was critically important as it is the family, rather than the individual, that is central in Indigenous cultures and communities. Indigenous knowledge includes the family as a unit of healing; participants’ entire families engage in care with them, including children, parents, and grandparents.

## 5. Discussion

Focus group participants shared experiences of poor-quality care at multiple levels of the healthcare system. The lack of effective care frequently resulted in negative health outcomes and experiences for the participant and/or their family members. Participants attributed the poor quality of care to a bias against Indigenous patients, including stereotyping, racism, and a lack of knowledge about Indigenous people. The experiences shared by participants in this study are supported by an extensive body of research that demonstrates inequitable health experiences and outcomes among diverse racial and ethnic groups compared to Non-Hispanic White patients [6,35,36,37,38,39,40,41,42]. The pathway from multilevel racism and discrimination to poor health outcomes and health disparities is presented in Figure 8. The figure was created using the published literature and the research findings from this study, and builds upon three previous models. As shown in the figure, institutional and interpersonal bias are shaped by broader societal bias and often operate in tandem with the biases and stereotypes at one level of society reinforcing and legitimizing the biases that exist at other levels.

Institutional biases and associated structural barriers contribute to inequitable access to social, educational, and material resources. Lack of quality health services and resources directly impacts health outcomes through the ability to receive quality care and critical health information and actualize recommendations [37,43]. There are also indirect effects on health in the form of stress and declining agency and trust in the healthcare system that result in decreased utilization of healthcare services, poor health outcomes, and increased disparities [37,38]. Both pathways, institutional and interpersonal, contribute to growing health disparities.

**Figure 8 ijerph-22-00445-f008:**
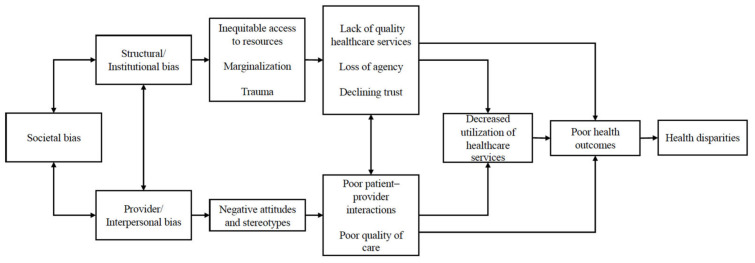
Conceptual framework for the impacts of institutional and interpersonal biases on health disparities. Adapted from Zestcott et al. (2016), Blair et al. (2011), and Lewis et al. (2021) [39,41,44] and informed by the literature cited in the Introduction and Study Results.

The results of this research provide yet another example of how negative assumptions and biases at both clinical provider and structural levels contribute to health disparities in minority populations [39,41,44]. Participants felt unheard and worried about receiving poor care, especially because they were Indigenous. They wished they could be full partners in decisions impacting their health. Poor clinical care in turn led to decreased utilization of services, poorer health outcomes, and increased disparities. Examples of the biased care patients encountered included inadequate assessments of problems or, even worse, a dismissal of symptoms, as well as intrusive questioning of sobriety and financial status.

Many people may assume that bias towards Indigenous people in a clinic setting today would be rather innocuous, resulting in an inconvenience or mildly hurt feelings. For this study, a clear red thread appeared, which was that the provision of bias-free care would create quality care, while biased care resulted in poor quality of care. This finding is supported by the participants’ lived experiences and a growing body of literature that confirms the negative health effects of bias against minorities in healthcare settings [45,46,47,48]. While the literature clearly demonstrates the relationship between the experiences of bias and racism in other ethnic minority groups, almost no literature exists for the Indigenous population that participated in this study. This study’s results demonstrate that bias has real consequences that reduce the quality of care and most likely lead to missed or incorrect diagnoses. These results support that bias plays a large role in Indigenous health outcomes, suggesting the need for more attention and research in this area.

Participants described a clear vision of an effective health encounter that elevates respectful interactions and uplifts Indigenous people and their health beliefs. To realize this vision, specific recommendations included interpersonal skills training for physicians, appropriate billing protocols, continuity of care, care that is free of stigma, skilled clinical decisions, integrated care, patient agency, and Indigenous culture and healing. Participants contrasted their goals and visions of positive and effective health encounters with their lived realities in which they received poor clinical care due to bias and racism. Discrimination did not always look blatant, such as with racial slurs, but was most often revealed through less than quality care, including limited assessments, inaccurate diagnoses, inaccurate treatment plans, and a lack of follow-up via treatment instructions, check-ins, etc. Figure 9 presents the experience of poor clinical care, where societal, institutional, and provider bias lead to poor patient–provider interactions, quality of care, and health outcomes, and persistent health disparities.

On the other hand, positive healthcare encounter (Figure 10) examples centered on the provision of culturally appropriate, comprehensive care from providers that were bias-free and who understood the basic cultural needs, validated their experiences, and utilized shared decision-making. Positive policies around structure were those that reduced barriers to care (administrative, physical, cultural, and clinical (Figure 4)), incorporated cultural traditions, and partnered with communities. In summary, participants described a clear vision of effective health encounters and systems that are bias-free, facilitate respectful encounters, and prioritize Indigenous people and Indigenous health beliefs.

### 5.1. Future Directions

This study illustrates the value of the patient’s voice in working to improve Indigenous health and the health of other health disparate communities. Lessons from this study can be incorporated into many settings. For example, it is important that the provision of culturally safe and appropriate care requires healthcare providers to have a deeper knowledge of the historical events that affect the health and well-being of their patients and communities, including understanding and appreciating health beliefs and values [49].

For Indigenous-specific care, healthcare providers must shift their communication style, beliefs, and behaviors [50]. Culturally appropriate and Indigenous-centered holistic healthcare must incorporate native staff, language, traditional healing practices, and a basic understanding of Indigenous values and beliefs [51]. The following are examples of a variety of different indigenous models of health and frameworks: Māori Te whare tapa whā [52] and MIHI models [53], the First Nations Mental Wellness Continuum Framework [54], the Alaska Native qasgiq model [55], Diné (Navajo) Hózhó Wellness Philosophy [56], and the Native Hawaiian conceptual model of NĀ Piko ‘Ekolu [57]. All of these health models were created or co-created with and for specific Indigenous communities.

What these models do not yet provide, however, is a practical integrated training curriculum to help healthcare providers and healthcare systems perform the work and make the changes that are needed to improve Indigenous health and Indigenous patient experiences. The findings of this project constitute a critical step in developing an Indigenous health toolkit that is responsive to the needs of Indigenous people [58]. The toolkit would incorporate many of the lessons learned from our focus groups and should be designed to be a practical training “toolkit” that can be used by both healthcare providers and healthcare systems.

### 5.2. Limitations

This study was conducted in the upper Midwest with 20 Lakota and Anishinaabe participants. Participants completed focus group interviews at the clinic that they were recruited from. It is important to consider that, while the location was convenient for participants, it was not neutral, which may have limited participants’ feelings of comfort in addressing challenges they experienced at that site. Future projects will work to ensure a neutral and convenient setting for participants.

Given the unique historical and current circumstances of each tribal nation, this study’s results cannot be generalized to other communities. However, the findings do highlight several central themes regarding bias, the availability of resources, and the importance of cultural knowledge and beliefs that are likely relevant to other Indigenous groups. Given the significant health disparities that exist in many Indigenous populations, it is imperative that more research be performed to document and understand Indigenous patients’ experiences and needs and to inform the development of tribe-specific recommendations. Specifically, the link between different types of bias and health disparities in minority groups highlights the need to better understand and address the interplay between institutional and interpersonal biases in healthcare, particularly in Indigenous populations, which are understudied [38,39,41,59,60].

## 6. Conclusions

Indigenous people experience significant barriers to quality healthcare [1,2,4,5,22]. These factors include institutional and interpersonal biases against Indigenous people influenced by an extensive history of colonization, trauma, marginalization, and exploitation. An improvement in the education of all Americans could help reduce societal level bias, but there is no systematic educational plan to do so. Additionally, many legislatures are becoming increasingly hostile to attempts to provide more accurate and comprehensive instruction on the history of groups who have been mistreated and disenfranchised. Therefore, it is important that healthcare training institutions take a more active role at the structural and provider levels to correct the false and harmful narratives about Indigenous people that have resulted in this group experiencing the worst health outcomes in the country. Training of healthcare professionals is feasible and effective at reducing bias and improving the care provided to Indigenous patients. Appropriately trained providers and administrators can improve care delivery, facilitate institutional policies that reflect the needs of Indigenous patients and their communities, and provide care reflective of Indigenous health beliefs. All these factors are critical for improving the health and quality of life of Indigenous people.

## Figures and Tables

**Figure 1 ijerph-22-00445-f001:**
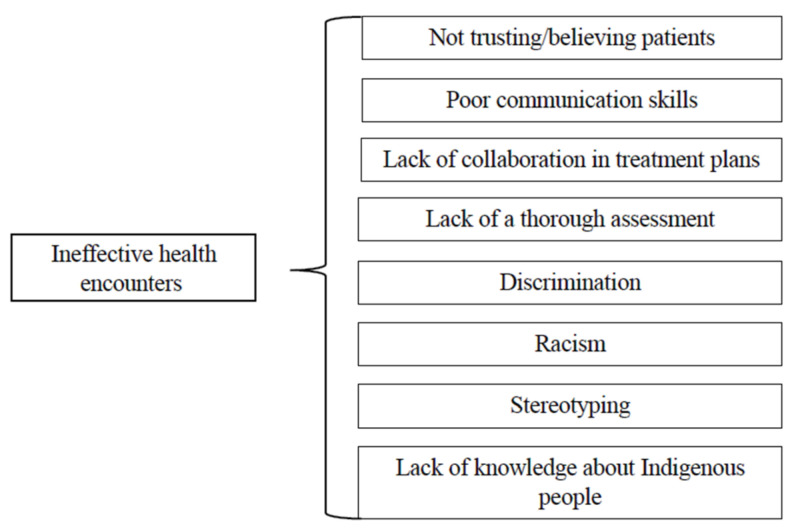
Ineffective health encounters: third-level themes.

**Figure 2 ijerph-22-00445-f002:**
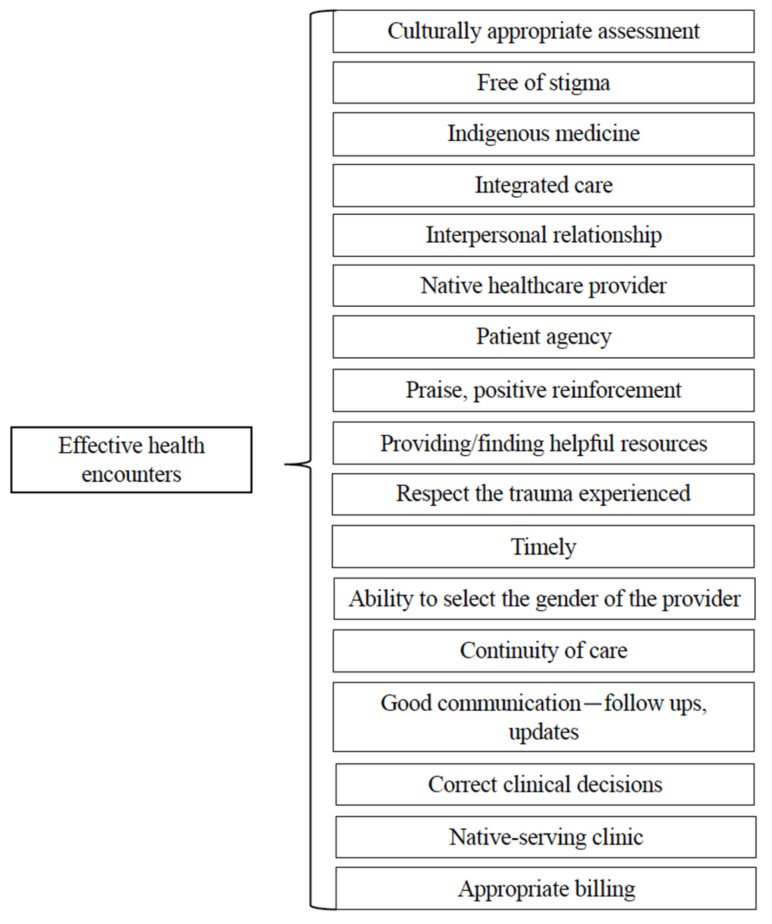
Effective healthcare encounters: third-level themes.

**Figure 3 ijerph-22-00445-f003:**
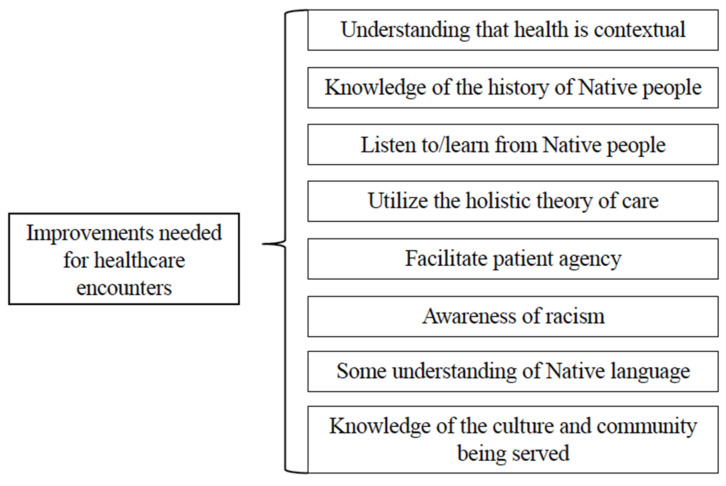
Third-level themes for improvements needed for healthcare encounters.

**Figure 4 ijerph-22-00445-f004:**
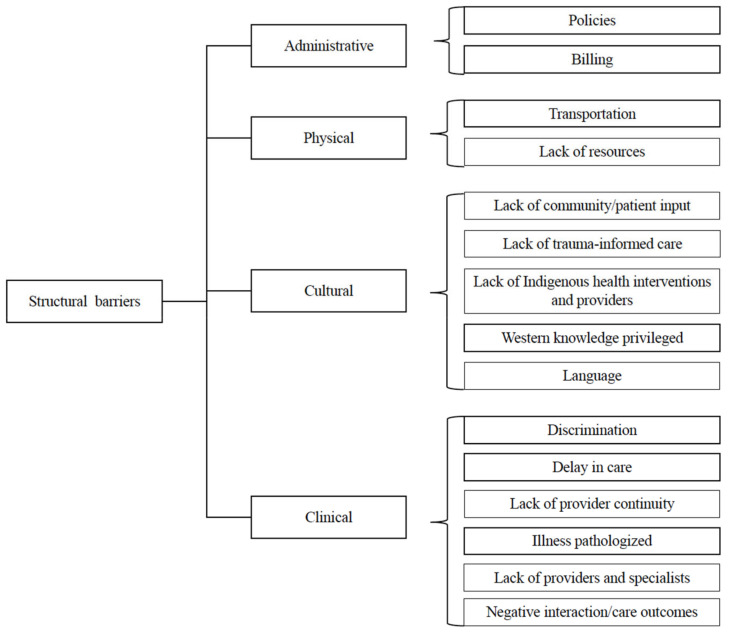
Third- and fourth-level themes related to structural barriers.

**Figure 5 ijerph-22-00445-f005:**
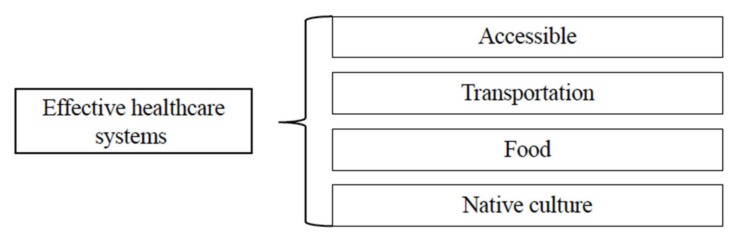
Third-level themes related to effective healthcare systems.

**Figure 6 ijerph-22-00445-f006:**
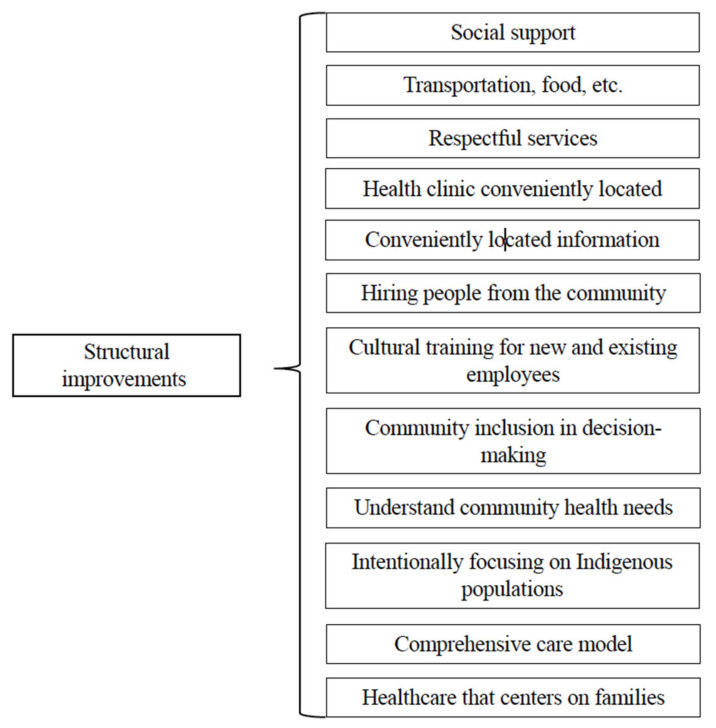
Third-level themes related to improvements needed for healthcare systems.

**Figure 7 ijerph-22-00445-f007:**
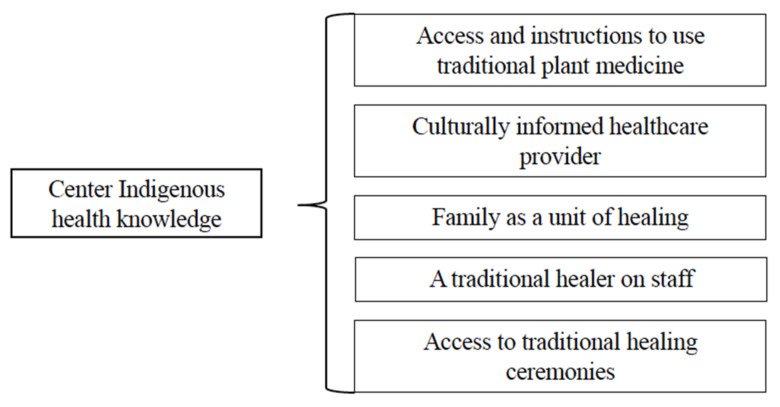
Second- and third-level themes related to Indigenous knowledge and beliefs.

**Figure 9 ijerph-22-00445-f009:**
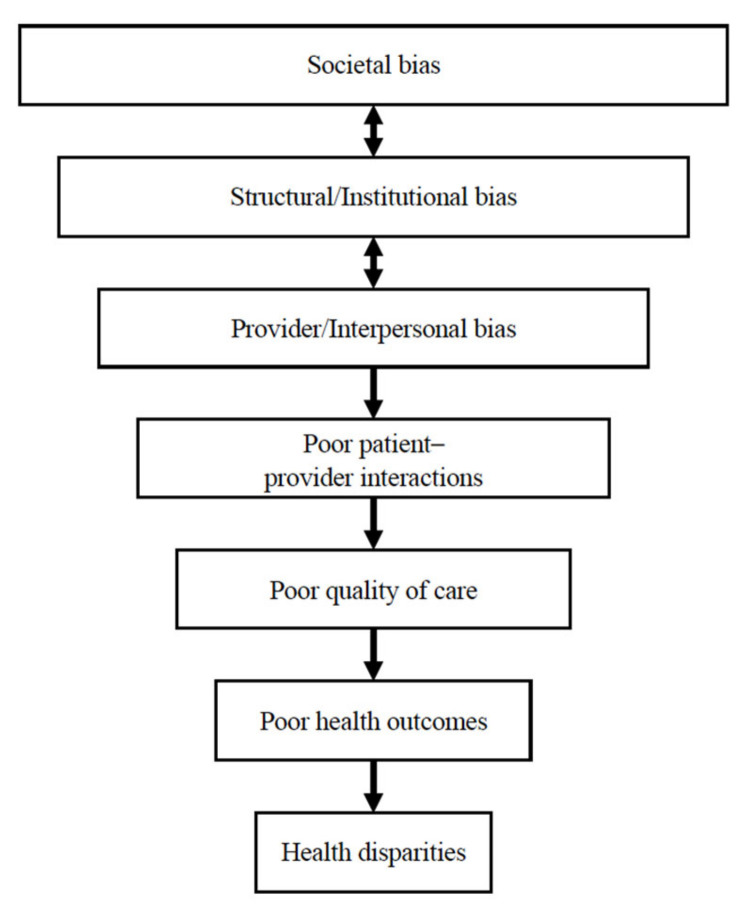
Poor-quality clinical care.

**Figure 10 ijerph-22-00445-f010:**
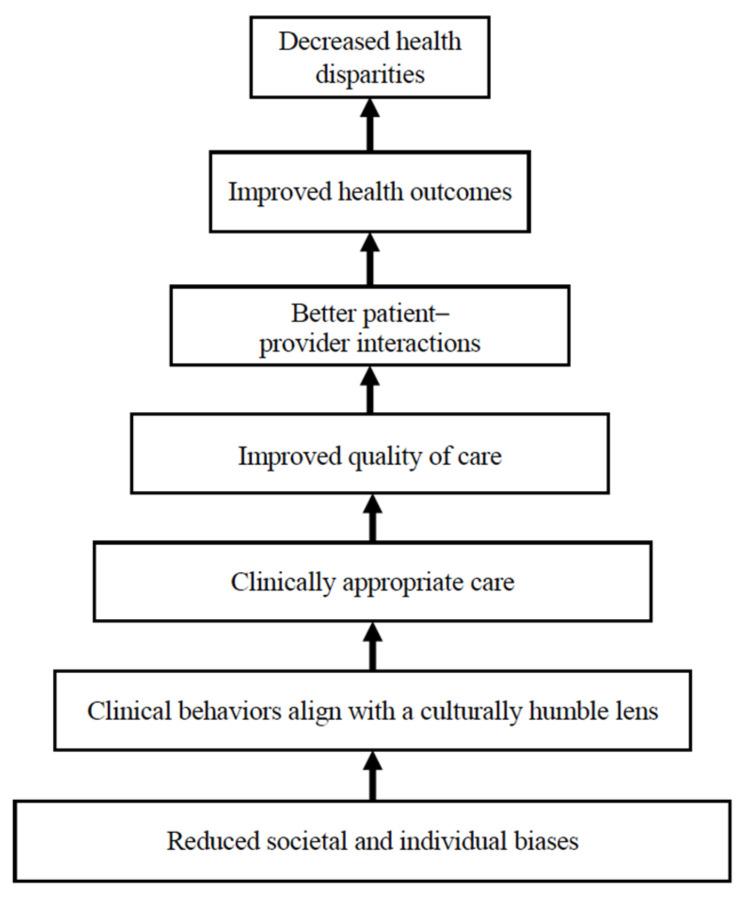
High-quality, culturally responsive care.

**Table 1 ijerph-22-00445-t001:** Participants’ demographics (*n* = 20).

	Mean	SD
Age (years)	46	12
Annual income	USD 11,709	USD 9746
	n	%
Gender		
Female	16	80%
Male	3	15%
Gender neutral	1	5%
Tribal group		
Anishinaabe	20	75%
Lakota	5	25%
Reported # of physical illnesses		
0	6	30%
1	10	50%
2	3	15%
4	1	5%
Reported # of emotional illnesses		
0	6	30%
1	9	45%
2	3	15%
3	2	10%
Reported substance use		
0	10	50%
1	6	30%
2	3	15%
3	1	5%
First language (n = 19)		
English	16	84%
Indigenous (Anishinaabe or Lakota)	3	16%
Language currently spoken (n = 19)		
English only	16	84%
English and Spanish	2	10%
English and Indigenous (Anishinaabe or Lakota)	1	1%

**Table 2 ijerph-22-00445-t002:** Participant reports regarding assumptions providers made about Indigenous patients.

Assumptions	Examples from Focus Groups
Behavior	Drug seeking
	Drug addict
	Alcoholic
Capabilities	Not knowledgeable
	Not smart
	Cannot make informed decisions
	Not capable parenting skills and knowledge
Responsibility	Not fiscally responsible
	Will not adhere to treatment

## Data Availability

Currently unavailable to the public.

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
