# Peer review of "“Understand the Way We Walk Our Life”: Indigenous Patients’ Experiences and Recommendations for Healthcare in the United States"

_ijerph, 2025, doi:10.3390/ijerph22030445_

Round 1

Reviewer 1 Report

Comments and Suggestions for Authors

Line 20-21: It appears that the work four is missing.

Line 197: Table 2 is labeled as Assumptions Providers Made about Indigenous patients. Isn't it really assumptions the research subjects reported as assumptions providers made? This may be true of other figures as well.

Line 406: Figure 7 is derived from other individuals' research. Would it be possible to create an accompanying figure that shows how Figure 7 would change if all of the information gathered in this study were implemented? Who wouldn't want that figure to use as a goal of this research?

Author Response

Reviewer 1 Comment Author Response (ML)
  Line 20-21: It appears that the work four is missing. Yes-thank you. The word "four" was added.

  Line 197: Table 2 is labeled as Assumptions Providers Made about Indigenous patients. Isn't it really assumptions the research subjects reported as assumptions providers made? This may be true of other figures as well. Yes, thank you again. We added "Participant reports of..."

  Line 406: Figure 7 is derived from other individuals' research. Would it be possible to create an accompanying figure that shows how Figure 7 would change if all of the information gathered in this study were implemented? Who wouldn't want that figure to use as a goal of this research? Thank you for this input. We will clarify this figure further in the text. This figure is a combination of data from the literature review, and from the study findings. We used this data and added to pre-exiting figures. 

Reviewer 2 Report

Comments and Suggestions for Authors

This was a well organized, timely, and scientifically sound paper. Qualitative interviews and capturing perspectives and dimensions of patient experiences are increasingly called for increasingly input in research in order to appropriately illustrate the healthcare and service gaps for Indigenous Peoples. The clarification of what appropriate cultural services looks like is also a key strength of this article that is a strong and positive contribution. I believe that while these issues have been addressed in the broader Indigenous health research, it is increasingly important to meaningfully address and advance directions, strengths, and persistent systemic barriers. 
The article is cohesive, concisely and clearly written, exploratory and easy to read, which is great considering the authors invested substantial effort into appropriately detailing different dimensions of Indigenous healthcare. Some very brief and minor notes:

The initial literature review and introduction has some slightly outdated research - while their messages continue to be true, there is ongoing research that reinforce the validity of these points, that has been published more recently, and, in fact, can be included to attest to the duration of healthcare service gaps in primary care. 

Secondly, and this is truly the tiniest detail, but on line 327 (document p. 10\20), that has possessive apostrophes that I believe are not grammatically necessary; it should read "the community, its needs and its values". it's would read "it is". 
My thanks to the authors for this strong work.

Author Response

Reviewer 2 Comment Author Response
  This was a well organized, timely, and scientifically sound paper. Qualitative interviews and capturing perspectives and dimensions of patient experiences are increasingly called for increasingly input in research in order to appropriately illustrate the healthcare and service gaps for Indigenous Peoples.  Thank you.
  The clarification of what appropriate cultural services looks like is also a key strength of this article that is a strong and positive contribution. I believe that while these issues have been addressed in the broader Indigenous health research, it is increasingly important to meaningfully address and advance directions, strengths, and persistent systemic barriers. Thank you.
  The article is cohesive, concisely and clearly written, exploratory and easy to read, which is great considering the authors invested substantial effort into appropriately detailing different dimensions of Indigenous healthcare. Some very brief and minor notes: Thank you.

1 The initial literature review and introduction has some slightly outdated research - while their messages continue to be true, there is ongoing research that reinforce the validity of these points, that has been published more recently, and, in fact, can be included to attest to the duration of healthcare service gaps in primary care. Literature review has been updated

2 Secondly, and this is truly the tiniest detail, but on line 327 (document p. 10\20), that has possessive apostrophes that I believe are not grammatically necessary; it should read "the community, its needs and its values". it's would read "it is". Edited as directed
  My thanks to the authors for this strong work.  

Reviewer 3 Report

Comments and Suggestions for Authors

This manuscript highlights the obstacles faced by Indigenous people in America in accessing culturally appropriate healthcare and offers suggestions for improving healthcare access. The manuscript is well-written, theoretically engaging, and compelling, enriched by vivid vignettes. However, certain areas require further elaboration to strengthen the study’s credibility and impact:

  1. Background Information and Context
    There is limited background information about the study setting, which is crucial for establishing ecological validity. For example, it is unclear what average distances Indigenous participants must travel to reach healthcare facilities. This information is essential to understanding the geographic and logistical barriers they face. Additionally, the manuscript suggests that many participants lack health insurance and are required to make upfront payments, indicating significant economic challenges. Providing detailed context about the socio-economic and geographic environment of the participants would enhance the study’s depth and validity.

  2. Ethical Considerations
    The manuscript lacks a discussion on the ethical dimensions of the study. Key questions remain unanswered, such as:

    • Where was ethical clearance for the study obtained?
    • How were participants’ consent and confidentiality ensured?
      Addressing these aspects is vital to uphold the integrity of the research process and to reassure readers of the study’s adherence to ethical standards.
  3. Challenges of Interviewing Patients During Healthcare Visits
    Interviewing healthcare patients at the point of service delivery presents unique challenges that could influence the findings. These challenges include:

    • Emotional and Physical Stress: Patients may be in pain, distress, or discomfort, affecting the quality of their responses.
    • Privacy and Confidentiality Concerns: The clinical setting may compromise participants' willingness to share personal or sensitive information.
    • Impact on Healthcare Delivery: Interviews could disrupt care provision or create time pressures for both patients and healthcare staff.
    • Potential for Social Desirability Bias: Participants might provide responses they believe are expected or socially acceptable.
    • Limited Reflection Time: Patients may not have sufficient time or mental bandwidth to reflect on their experiences deeply.
    • Influence of the Healthcare Environment: The clinical setting may induce stress or inhibit honest communication.
    • Ethical and Consent Challenges: Ensuring voluntary and informed consent in a healthcare setting requires careful planning and sensitivity.

Addressing these challenges in the methodology and discussion sections would significantly enhance the study's reliability and provide a more nuanced interpretation of the findings. Including these details would make the manuscript more robust, contributing to its theoretical and practical significance.

Author Response

Reviewer 3 Comment Author Response
  This manuscript highlights the obstacles faced by Indigenous people in America in accessing culturally appropriate healthcare and offers suggestions for improving healthcare access. The manuscript is well-written, theoretically engaging, and compelling, enriched by vivid vignettes. However, certain areas require further elaboration to strengthen the study’s credibility and impact:  

  1. Background Information and Context  
  There is limited background information about the study setting, which is crucial for establishing ecological validity. For example, it is unclear what average distances Indigenous participants must travel to reach healthcare facilities. This information is essential to understanding the geographic and logistical barriers they face. Additionally, the manuscript suggests that many participants lack health insurance and are required to make upfront payments, indicating significant economic challenges. Providing detailed context about the socio-economic and geographic environment of the participants would enhance the study’s depth and validity. While we did not collect data in distance to clinic or health insurance status for this study in this study, we added additional contextual information about the study setting.

  2. Ethical Considerations  
  The manuscript lacks a discussion on the ethical dimensions of the study. Key questions remain unanswered, such as:  

  Where was ethical clearance for the study obtained? Ethics and ethical clearance were added and are discussed on lines Line 72-110
  How were participants’ consent and confidentiality ensured? Informed consent protocol was added starting on line 103.
  Addressing these aspects is vital to uphold the integrity of the research process and to reassure readers of the study’s adherence to ethical standards.  

  3. Challenges of Interviewing Patients During Healthcare Visits  
  Interviewing healthcare patients at the point of service delivery presents unique challenges that could influence the findings. These challenges include: Ethics of informed consent and focus group sessions were added and discussed on lines Line 72-110

  Emotional and Physical Stress: Patients may be in pain, distress, or discomfort, affecting the quality of their responses. Participants did not attend focus group sessions on the day of their appointment. They were recruited on the day of their appointment and scheduled for focus groups at a later date.  This section is rewritten for clarification.
  Privacy and Confidentiality Concerns: The clinical setting may compromise participants' willingness to share personal or sensitive information. While patients were not seen in a clinical area of the building, we will add this to the limitations section starting on line 504
  Impact on Healthcare Delivery: Interviews could disrupt care provision or create time pressures for both patients and healthcare staff. Interviews were held after clinic hours in non-clinical rooms. No staff reported disruptions 
  Potential for Social Desirability Bias: Participants might provide responses they believe are expected or socially acceptable. Given that social desirability bias is possible for all focus group research, as well as all self-report research methods, we do not think it is an important addition in the methods section
  Limited Reflection Time: Patients may not have sufficient time or mental bandwidth to reflect on their experiences deeply. Participants completed focus groups at a different day than their appointment
  Influence of the Healthcare Environment: The clinical setting may induce stress or inhibit honest communication. We will add this to the limitations section starting on line 504
  Ethical and Consent Challenges: Ensuring voluntary and informed consent in a healthcare setting requires careful planning and sensitivity. Yes, Lines 82-95 discuss the collaboration that took place to launch this study

  Addressing these challenges in the methodology and discussion sections would significantly enhance the study's reliability and provide a more nuanced interpretation of the findings. Including these details would make the manuscript more robust, contributing to its theoretical and practical significance. We edited to address this recommendation within our methodology and discussion sections.

Reviewer 4 Report

Comments and Suggestions for Authors

Overall: This manuscript discussed a really important narrative in Indigenous health. I especially
appreciate your level of importance to Indigenous Research Methods and the use of the “red
thread” to connect results. I have provided multiple considerations for you in hopes to
strengthen your paper and not by any means as criticism to the work.
Abstract:
● Lines 14-15 are confusing as written, “Indigenous communities experiences worsened
quality of care compared to non-Indigenous patients.” I would recommend, “Medical
experiences regarding quality of care for Indigenous communities has worsened
compared to non-Indigenous patients.”
● Lines 20-21 need editing- “Twenty Indigenous patients participated in one of focus
groups regarding their experiences of healthcare systems.”
● There is an underscore before the results and conclusions. Remove.
● Lines 24-26 need editing- “Participants in this study highlighted that biased care results
in poor quality of care and that there are actionable steps that providers and systems of
healthcare can take to reduce bias within healthcare systems.” It is too wordy. Consider
splitting.
Introduction:
● The first sentence is very long (lines 33-36). Consider splitting the sentence for
increased clarity.
● Lines 46-47: “Bias, racism, and disrepect for health beliefs naturally results in the
avoidance of care which is linked to worse health outcomes for Indigenous people.
○ Disrespect is misspelled
○ I would say, “...can result in the avoidance..”
● There are multiple places were there should be a space between the end of the sentence
and the “(“ for the citation. Lines 37, 43, 45, 58, and 60.
Methods:
● Insert space between the end of the sentence and the “(“ for the citation. Lines 84, 88,
91.
● Line 79. You already defined the abbreviation of IRM in line 72, no need to repeat here.
● Line 89 used abbreviations PI and CEO without defining.
● Line 99. Did you do this via electronic survey or paper survey?

● Line 103. Was the 3rd party a person or technology?
● Please provide the initials of who led the focus groups and if other authors were involved
in the Data Collection section.
● How long were the focus groups? When were these interviews conducted? Please add
details in the Data Collection section.
● Line 106. I think this whole section goes to results.
● Line 115-116. I would say mental health condition versus behavioral health illness per
your definitions list after the term.
● Line 117. Add “disorders” after substance abuse.
● Line 136. Provide initials of team members who were included in this process.
● Lines 138-144. Does this follow a specific qualitative analysis process? This process
should be cited.
● Line 143- I would call these seven sub themes and three main themes.
Results:
● Line 146- again, add sub themes
● Lines 167-170 note that the figure shows, “An overview of the ineffective healthcare
encounter themes can be found in Figure 1.” then the title of the figure says, “Ineffective
health encounter sub-themes.” Align language here. I would actually further define these
words– I don’t think they are subthemes, instead something more descriptive like,
“observed and/or experienced interactions leading to ineffective healthcare encounters”
● Line 181- would you like to capitalize Native here?
● Lines 215 and 238- remove period within the parentheses
● Figure 2. I would say the same point as above– that long list of positive encounters are
not sub-themes, but instead “observed and/or experienced interactions leading to
effective healthcare encounters”
● Line 232. “Improved” should be “improving”
● Figure 3. Switch subthemes to suggestions
● Line 262- would say subthemes, not themes.

● Figure 4. Just subthemes here
● Line 277- “and provider training”.
● Figure 5. Suggestions versus subthemes here.
● Lines 360 and 367– this should be figure 6. You already have figure 5 in line 325.
● Figure 6 should be figure 7.
Discussion:
● Overall, the formatting of the discussion needs more integration. The research should be
woven into the literature outcomes– it reads a little clunky as written. Suggestions are
below.
● Lines 398-399. This sentence should be rephrased. I am not sure that the phrasing of
“unequal distribution of health outcomes” makes sense. Distribution is the wrong word
there.
● In the first paragraph, add a sentence or two about how your research contributes to this
understanding. A little summary of how your project ties into this larger body of work
would be helpful here.
● Same with paragraph 2 (lines 411-419). How does your project contribute to this work?
The reader wants to know what you’re contributing to the literature.
● Line 429 is a powerful sentence. Can that be supported by other literature? Or integrate
information from the second paragraph to make these two paragraphs more cohesive?
● I think you need to accentuate the theme of your red thread in lines 432-444. Why is this
important? What does other literature show about the importance of the statement,
“...bias free care would create quality care, while biased care results in poor quality of
care”. This is not a unique conclusion, so it needs to be tied to other research. You are
describing it more in this paragraph, but I think there needs to be more support from
other research and a strong theme alignment.
● Figure 7 should be figure 8.
● Figure 8 should be figure 9
● Paragraph 5, Lines 449-456 needs to provide more detail about the systemic barriers that
are present, related to your findings.
● Figure 9 should be figure 10 .

Author Response

Reviewer 4 Comment Author Response - IB
  Overall: This manuscript discussed a really important narrative in Indigenous health. I especially appreciate your level of importance to Indigenous Research Methods and the use of the “red thread” to connect results. I have provided multiple considerations for you in hopes to strengthen your paper and not by any means as criticism to the work.  
  Abstract:  
  ● Lines 14-15 are confusing as written, “Indigenous communities experiences worsened Thank you for this suggestion, we have edited the text
  quality of care compared to non-Indigenous patients.” I would recommend, “Medical  
  experiences regarding quality of care for Indigenous communities has worsened  
  compared to non-Indigenous patients.”  
  ● Lines 20-21 need editing- “Twenty Indigenous patients participated in one of focus Thank you noting this, the underscore has been removed and the word "four" added
  groups regarding their experiences of healthcare systems.”  
  ● There is an underscore before the results and conclusions. Remove.  
  ● Lines 24-26 need editing- “Participants in this study highlighted that biased care results This is a helpful suggestion,  the sentence has been edited for clarity
  in poor quality of care and that there are actionable steps that providers and systems of  
  healthcare can take to reduce bias within healthcare systems.” It is too wordy. Consider  
  splitting.  
  Introduction:  
  ● The first sentence is very long (lines 33-36). Consider splitting the sentence for This sentence has been edited for clarity
  increased clarity.  
  ● Lines 46-47: “Bias, racism, and disrepect for health beliefs naturally results in the The noted typo has been corrected and suggested edit made
  avoidance of care which is linked to worse health outcomes for Indigenous people.  
  ○ Disrespect is misspelled  
  ○ I would say, “...can result in the avoidance..”  
  ● There are multiple places were there should be a space between the end of the sentence Thank you for noting this, spaces have been added
  and the “(“ for the citation. Lines 37, 43, 45, 58, and 60.  
  Methods:  
  ● Insert space between the end of the sentence and the “(“ for the citation. Lines 84, 88, Thank you for noting this, spaces have been added
  91  
  ● Line 79. You already defined the abbreviation of IRM in line 72, no need to repeat here. The repeat of Indigenous Research Methodology has been deleted
  ● Line 89 used abbreviations PI and CEO without defining. PI and CEO have been defined
  ● Line 99. Did you do this via electronic survey or paper survey? It was a paper survey

  ● Line 103. Was the 3rd party a person or technology? A third party person deidentified and transcribed the recordings
  ● Please provide the initials of who led the focus groups and if other authors were involved The focus groups were conducted by trained graduate students who used a facilitator's guide. This detail has been added to the data collection section
  in the Data Collection section.  
  ● How long were the focus groups? When were these interviews conducted? Please add Dates and lengths added
  details in the Data Collection section.  
  ● Line 106. I think this whole section goes to results. Agreed. This section has been moved to the begining of the Results section
  ● Line 115-116. I would say mental health condition versus behavioral health illness per We have changed to 'behavioral health illness' to 'mental health diagnosis'
  your definitions list after the term.  
  ● Line 117. Add “disorders” after substance abuse. " disorders" has been added
  ● Line 136. Provide initials of team members who were included in this process. Initials have been added
  ● Lines 138-144. Does this follow a specific qualitative analysis process? This process Yes, it follows the qualitative analytical process described in the paragrpah before this section. Citations have been added to clarify the connection between the two paragraphs
  should be cited.  
  ● Line 143- I would call these seven sub themes and three main themes. Thank you for noting this. The language that specifies the number of main and subthemes has been removed since we deteremined that the number of themes belonged in the Results
  Results:  
  ● Line 146- again, add sub themes Thank you, the language of this sentence has been edited to reflect three main themes and seven subthemes
  ● Lines 167-170 note that the figure shows, “An overview of the ineffective healthcare The sentence noted has been deleted as it was deteremined to not be necessary. We have added language near the begining of the Results section explaining that the themes are organized into three distinct levels. What the reviewer refers to as "observed and/or experieced interactions" is actually the third level of themes. We realize this may not have been clear. The description of the results throughout the rest of the Results section aligns with this terminology and organization. 
  encounter themes can be found in Figure 1.” then the title of the figure says, “Ineffective  
  health encounter sub-themes.” Align language here. I would actually further define these  
  words– I don’t think they are subthemes, instead something more descriptive like,  
  “observed and/or experienced interactions leading to ineffective healthcare encounters”  
  ● Line 181- would you like to capitalize Native here? Yes, Native have been capitlaized in this sentence
  ● Lines 215 and 238- remove period within the parentheses The period within the parentheses has been removed
  ● Figure 2. I would say the same point as above– that long list of positive encounters are What the reviewer refers to as "observed and/or experieced interactions" is actually a third level of themes. We realize this may not have been clear. Language has been added to the results section to clarify.
  not sub-themes, but instead “observed and/or experienced interactions leading to  
  effective healthcare encounters”  
  ● Line 232. “Improved” should be “improving” "improved" is the word we intended to use, it has not been changed
  ● Figure 3. Switch subthemes to suggestions The title description has been edited to align with the terminology and organization of the rest of the Results section
  ● Line 262- would say subthemes, not themes. This sentence has been edited to align with the terminology and organization of the rest of the Results section

  ● Figure 4. Just subthemes here The title description has been edited to align with the terminology and organization of the rest of the Results section
  ● Line 277- “and provider training”. "and" has been added 
  ● Figure 5. Suggestions versus subthemes here. The title description has been edited to align with the terminology and organization of the rest of the Results section
  ● Lines 360 and 367– this should be figure 6. You already have figure 5 in line 325. Thank you for noting this error. The number has been changed.
  ● Figure 6 should be figure 7. Thank you for noting this error. The number has been changed.
  Discussion:  
  ● Overall, the formatting of the discussion needs more integration. The research should be  
  woven into the literature outcomes– it reads a little clunky as written. Suggestions are  
  below.  
  ● Lines 398-399. This sentence should be rephrased. I am not sure that the phrasing of This sentence was edited and noted phrasing was deleted
  “unequal distribution of health outcomes” makes sense. Distribution is the wrong word  
  there.  
  ● In the first paragraph, add a sentence or two about how your research contributes to this Thank you, we agree with this observation. We have added several sentences at the begining of this paragraph that tie in the research finds
  understanding. A little summary of how your project ties into this larger body of work  
  would be helpful here.  
  ● Same with paragraph 2 (lines 411-419). How does your project contribute to this work? Edits have been had to this paragraph and the one below it to better describe how the research contributes to the literature
  The reader wants to know what you’re contributing to the literature.  
  ● Line 429 is a powerful sentence. Can that be supported by other literature? Or integrate Line 429 describes the experiences of focus group participants "Examples of biased care patients encountered included inadequate assessment of problems or even worse, dismissal of symptoms as well as excessive questioning of sobriety and financial status". We are unclear why there would be a need to support this sentence with cited literature. Maybe the reviewer meant to refer to a different line number?
  information from the second paragraph to make these two paragraphs more cohesive?  
  ● I think you need to accentuate the theme of your red thread in lines 432-444. Why is this Thank you for this suggestion. We have made edits and added citations to strengthen the argument in this paragraph
  important? What does other literature show about the importance of the statement,  
  “...bias free care would create quality care, while biased care results in poor quality of  
  care”. This is not a unique conclusion, so it needs to be tied to other research. You are  
  describing it more in this paragraph, but I think there needs to be more support from  
  other research and a strong theme alignment.  
  ● Figure 7 should be figure 8. Thank you for noting this error. The number has been changed.
  ● Figure 8 should be figure 9 Thank you for noting this error. The number has been changed.
  ● Paragraph 5, Lines 449-456 needs to provide more detail about the systemic barriers that A reference for the Results section where the barriers are described in detail has been added 
  are present, related to your findings.  
  ● Figure 9 should be figure 10 . Thank you for noting this error. The number has been changed.